EMBO
Molecular Medicine

# Radiation combined with macrophage depletion promotes adaptive immunity and potentiates checkpoint blockade

Keaton I Jones[1] (iD), Jiske Tiersma[1,2], Arseniy E Yuzhalin[1], Alex N Gordon-Weeks[3], Jon Buzzelli[1], Jae Hong Im[1] & Ruth J Muschel[1,*] (iD)

## Abstract

Emerging evidence suggests a role for radiation in eliciting anti-tumour immunity. We aimed to investigate the role of macrophages in modulating the immune response to radiation. Irradiation to murine tumours generated from colorectal (MC38) and pancreatic (KPC) cell lines induced colony-stimulating factor 1 (CSF-1). Coincident with the elevation in CSF-1, macrophages increased in tumours, peaking 5 days following irradiation. These tumour-associated macrophages (TAMs) were skewed towards an immunosuppressive phenotype. Macrophage depletion via anti-CSF (aCSF) reduced macrophage numbers, yet only achieved tumour growth delay when combined with radiation. The tumour growth delay from aCSF after radiation was abrogated by depletion of CD8 T cells. There was enhanced recognition of tumour cell antigens by T cells isolated from irradiated tumours, consistent with increased antigen priming. The addition of anti-PD-L1 (aPD-L1) resulted in improved tumour suppression and even regression in some tumours. In summary, we show that adaptive immunity induced by radiation is limited by the recruitment of highly immunosuppressive macrophages. Macrophage depletion partly reduced immunosuppression, but additional treatment with anti-PD-L1 was required to achieve tumour regression.

**Keywords** immunosuppression; immunotherapy; macrophage; radiation
**Subject Categories** Cancer; Immunology

## Introduction

Over half of patients with cancer receive radiotherapy at some point during the course of their treatment (Baskar *et al*, 2012). The principal effect of radiation results from irreparable DNA damage. However, more recently it has become apparent that radiation has important modulatory effects on the immune response to the tumour. These are both immunostimulatory and immunosuppressive.

Immunostimulatory effects arise from increased tumour peptide availability along with increased expression of MHC class I proteins on the irradiated cancer cells that allow greater access for antigen presentation (Reits *et al*, 2006; Wan *et al*, 2012; Rudqvist *et al*, 2018). Damaged tumour cells release damage-associated molecular patterns (DAMPs) that stimulate an immune response, including enhanced recruitment and activity of antigen-presenting cells (Schaue & McBride, 2010). These factors can lead to tumour-specific adaptive immunity. Despite the potential for radiation to stimulate anti-tumour immunity, an effective response often fails to be generated due to immune escape through mechanisms including the expression of checkpoint molecules, T-cell exhaustion and generation of highly suppressive microenvironments through recruitment of specific subsets of myeloid cells (Vatner & Formenti, 2015; Zarour, 2016). Further elucidation of these factors contributing to immune resistance is imperative if the full potential of radiotherapy to potentiate the immune response is to be realised.

Tumour-associated macrophages (TAMs) are an abundant myeloid population present within the stromal compartment of many solid tumours. They are notable for their functional plasticity, allowing differentiation into a range of phenotypes. Under normal physiological conditions, macrophages mediate an acute pro-inflammatory response following tissue injury. These classically activated macrophages have been labelled as "M1", analogous to the $T_h1$ immune response, and are generally considered to exert anti-tumourigenic effects (Mantovani *et al*, 2002). At the other end of the polarisation spectrum, alternatively activated "M2" macrophages are generated during the later phases of healing after tissue injury. These macrophages can promote angiogenesis, extracellular matrix deposition and proliferation, secrete immunosuppressive cytokines and are generally considered to be pro-tumourigenic. Evidence for the role of macrophages in cancer is largely limited to the non-irradiated tumour setting. The effects of radiation on the

1   Department of Oncology, CRUK/MRC Oxford Institute for Radiation Oncology, Churchill Hospital, University of Oxford, Oxford, UK
2   Department of Medical Oncology, University Medical Centre Groningen, University of Groningen, Groningen, The Netherlands
3   Nuffield Department of Surgical Sciences, John Radcliffe Hospital, University of Oxford, Oxford, UK
    *Corresponding author. Tel: +44 (0)1865 225847; Fax: +44 (0)1865 617355; E-mail: ruth.muschel@oncology.ox.ac.uk

recruitment and phenotype of tumour-associated macrophages are less well reported. We aimed to determine the effect of radiation on macrophage recruitment and polarisation, and the role this population plays in the irradiated tumour microenvironment.

Here, we show that radiation stimulated a potential immune response that was balanced by increased numbers of immunosuppressive macrophages. Macrophage recruitment was promoted by radiation-induced upregulation of CSF-1 by tumour cells and was reversed by the administration of anti-CSF antibody (aCSF). We asked whether aCSF would enable an effective immune response. aCSF therapy resulted in macrophage depletion in naïve and irradiated tumours, but was associated with a CD8 T-cell-dependent anti-tumour response only when augmented by radiation-induced systemic tumour antigen priming. However, the induction of an immune response was still modest. Since surface PD-L1 on tumour cells was upregulated following radiation, the potential for robust and lasting anti-tumour immunity was still thwarted. The addition of an anti-PD-L1 antibody (aPD-L1) to aCSF resulted in improved tumour suppression and even regression in a highly resistant murine pancreatic cancer model. These data suggest that immunosuppressive macrophages limit radiation-induced adaptive immunity. Furthermore, macrophage depletion may play a role in immune checkpoint blockade-resistant tumours.

# Results

## Colony-stimulating factor 1 (CSF-1) is stimulated by irradiation of tumours

Irradiation of MC38 cells in culture with a single-dose 10 Gy irradiation (IR) induced expression of a variety of cytokines (Fig EV1A). Of those cytokines known to recruit myeloid cells after radiation, only CSF-1 was significantly elevated (Fig 1A). CSF-1 gene expression was significantly increased in MC38 cells at 24 h and in KPC cells at 48 h (Fig 1B), with elevated levels of CSF-1 protein in the media at 72 h as measured by ELISA (Fig 1C). *In vivo*, serum from mice bearing KPC tumours had elevated CSF-1 compared to naïve mice, whilst serum CSF-1 in mice bearing MC38 tumours was not elevated. However, after a single dose of 10 Gy to the tumours derived from both cell lines, CSF-1 was transiently elevated (Fig 1D).

In keeping with the increased levels of CSF-1, there was increased infiltration of macrophages in tumours (CD11b$^+$F480$^+$) within 48 h of single-dose radiation in both MC38 and KPC tumours (Fig 1E and F). The relative increase in TAMs persisted for 13 days in MC38 tumours, eventually returning to levels comparable to unirradiated controls after tumour regrowth resumed. Tumour sections collected 5 days following IR showed a dense infiltrate of CD68$^+$ macrophages (Fig 1G and H). We characterised some of the myeloid and lymphocytic populations in the tumour infiltrates (Fig 1I and J). There was also a significant increase in the relative number of CD45$^+$ cells in both types of tumour after radiation. These CD45$^+$ cells were predominantly myeloid cells, including macrophages, myeloid-derived suppressor cells and neutrophils. Lymphocytes were a minority of the immune infiltrates and remained largely unchanged following IR (Fig 1I and J). These results confirm that IR is associated with a relative increase in the myeloid compartment, including a significant, transient increase in TAMs.

## Macrophages recruited after irradiation display pro-tumourigenic markers

To define the activation status of the TAMs, we analysed expression of iNOS and CD206, as representative of M1 and M2 polarisation, respectively, by flow cytometry. The percentages of macrophages expressing iNOS, an inflammatory or M1 marker, decreased in MC38 tumours after IR, but increased in KPC tumours, not exceeding 30% (Fig 2A). Notably, the iNOS signal in TAMs present in MC38 tumours was less bright than that observed in KPC TAMs. This was highlighted by a prominent peak of iNOS$^{hi}$ cells in KPC TAMs compared to a shift from the isotype signal seen in MC38 TAMs (Fig 2A). Percentages of TAMs with the M2 marker CD206 decreased from 56 to 39% in MC38 tumours with radiation, but remained constant in KPC (31 vs. 34% Fig 2B). TAMs were consistently at several fold higher amounts in KPC tumours than in MC38 tumours (Fig 2C). However, because of the increased numbers of TAMs, this amounted to an overall increase in TAMs more polarised towards the M2 spectrum in both tumour types (Fig 2D and E). This resulted in a trend towards decreased M1:M2 ratio in both groups (Fig 2F). The gene expression profiles of isolated TAMs with and without radiation were compared (Fig 2G and H). The patterns from the macrophages from MC38 and KPC tumours after radiation were not identical; however, markers of immune suppression were generally higher in both groups. TAMs from naïve and irradiated MC38 tumours were suppressive, based on a T-cell suppressive assay (Fig 2I). TAMs from irradiated KPC tumours were also effective at suppressing T-cell proliferation, but not those from naïve tumours (Fig 2J).

To investigate whether tumour cell conditioning alone could be responsible for macrophage polarisation, we co-cultured BMDMs with naïve and irradiated tumour cells. Culture with irradiated cells induced significant increases in CD206 expression on BMDM, comparable to TAMs (Fig EV2A and B). Gene expression in MC38 co-cultured macrophages largely resembled that of TAMs (Fig EV2C), although expression of some inflammatory markers was increased in the KPC co-culture (Fig EV2C and D). The BMDM generated after co-culture with either of the tumour cell types had a significantly increased capacity to suppress T-cell proliferation (Fig EV2E and F).

## Anti-CSF therapy delays tumour growth following irradiation

Because radiation induced CSF1 in these tumours, we determined the effect of an anti-CSF1 antibody (aCSF) on TAMs and tumour growth delay after radiation (Fig 3A). Five days following radiation, TAM numbers were significantly reduced in aCSF-treated mice (Fig 3B and C). aCSF did not alter the growth of either MC38 or KPC tumours despite reduction in TAMs. Irradiation of tumours with 10 Gy of gamma rays resulted in a growth delay in both models (Fig 3D and E), which was approximately doubled by the addition of aCSF. aCSF did not affect clonogenic capacity of MC38 or KPC cells with or without radiation (Fig EV3A and B).

The fold reduction in macrophages was comparable between MC38 and KPC tumours after aCSF treatment, though KPC tumours

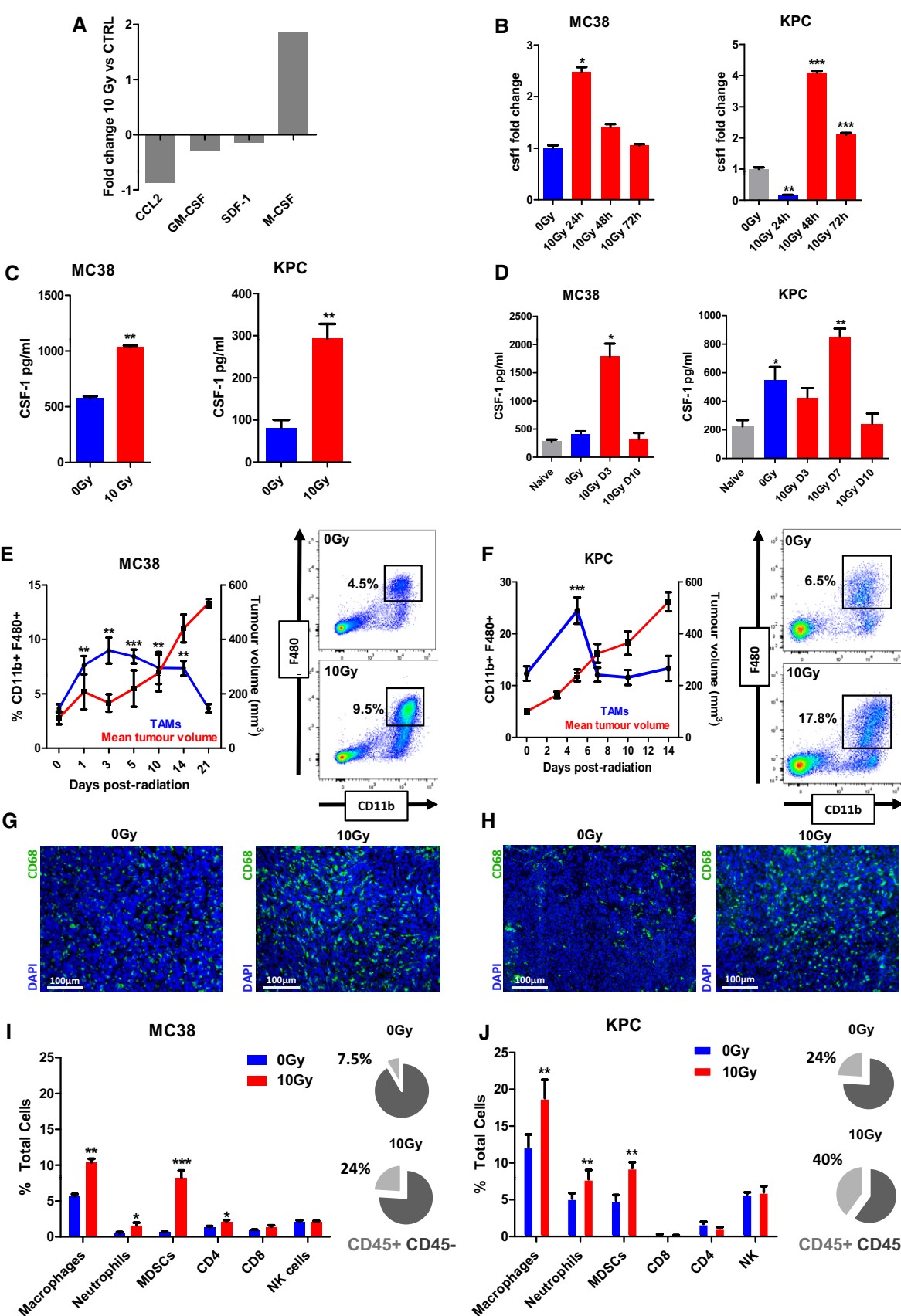

**Figure 1.**

**Figure 1.  Colony-stimulating factor 1 (CSF-1) and macrophage percentages increase in response to irradiation.**

A   MC38 cells in tissue culture were treated with mock (0 Gy) and 10 Gy irradiation. Conditioned medium (CM) was collected after 48 h and assayed for the indicated cytokines. Fold changes in the amounts of the cytokines are shown.

B   CSF-1 mRNA expression was measured in MC38 and KPC cells exposed to 10 Gy IR compared with mock-irradiated cells. Cells were harvested at 24, 48 and 72 h after irradiation, and RNA expression was analysed by RT–qPCR. Data are presented as mean ± SEM and analysed by Kruskal–Wallis test with Dunn's multiple comparisons test ($n = 3$).

C   CSF-1 protein (pg/mg total protein) in CM collected from MC38 and KPC cells 48 h after exposure to 10 Gy IR compared to mock-irradiated cells. Data are presented as mean ± SEM and analysed by Mann–Witney test ($n = 3$).

D   CSF-1 protein (pg/mg total protein) measured by ELISA in the sera of naïve mice, mice bearing mock-irradiated tumours and mice bearing irradiated tumours analysed at time points as indicated ($n = 4$/group). Data are presented as mean ± SEM and analysed by Kruskal–Wallis test with Dunn's multiple comparisons test.

E, F   MC38 (E) and KPC (F) tumours were irradiated with 10 Gy. Average tumour volume (red line) is shown with mean TAM infiltrate (blue line) for each time point. For TAM quantification, tumours were disaggregated and CD11b⁺/F480⁺ TAMs identified by flow cytometry. Data are presented as mean ± SEM for TAMs and SEM for tumour volume ($n = 6$). Data are presented as mean ± SEM and analysed by Kruskal–Wallis test with Dunn's multiple comparisons test.

G, H   Immunofluorescent staining of MC38 (G) and KPC (H) tumour sections; blue = DAPI, green = CD68 (TAMs).

I, J   Flow cytometric analysis of immune cell populations within MC38 (I) and KPC (J) tumours 5 days following 10 Gy IR compared to mock-irradiated tumours. Macrophages (CD11b⁺F480⁺), neutrophils (CD11b⁺Ly6G⁺), myeloid-derived suppressor cells (CD11b⁺Gr1⁺), CD8 T cells (CD45⁺CD3⁺CD8⁺), CD4 T cells (CD45⁺CD3⁺CD4⁺), and natural killer cells (CD45⁺NK1.1⁺) were identified. Pie charts represent the proportion of CD45⁺ leucocytes out of the total cells. Data are presented as mean ± SEM and analysed by unpaired *t*-test ($n = 3$).

Data information: *$P < 0.05$, **$P < 0.01$, ***$P < 0.001$.

generally had substantially higher overall TAM numbers (Fig 3B and C). Following aCSF treatment, a resistant population remained in both tumour models with or without radiation. This population had a consistent reduction in CD206^hi "M2" TAMs in both tumour models (Fig 3F). Changes in iNOS^hi "M1" TAMs were variable (Fig 3G). aCSF given with radiation led to an increase in iNOS^hi TAMs in MC38 tumours and no change in KPC tumours. Gene expression in TAMs isolated from tumours treated with combination IR and aCSF revealed a general trend towards greater expression of pro-inflammatory genes (Fig 3H and I), with increases in iNOS, interleukin-1A and B and a reduction in arginase, CCL2 and IL-10. Taken together, these data confirm that aCSF therapy effectively depletes TAMs following irradiation and is associated with repolarisation towards a more pro-inflammatory pattern of gene expression.

**Macrophage-depleted tumours are infiltrated by cytotoxic CD8 T lymphocytes**

The presence of CD8 T cells is a reflection of the extent of an anti-tumour immune response. In MC38 tumours, the decrease in TAMs following aCSF was associated with a relative increase in CD8-positive T lymphocytes (Fig 4A). KPC tumours had very few lymphocytes, almost 10-fold less than MC38 tumours. These findings are in line with existing reports, which similarly found pancreatic tumours to contain very few CD8 T cells. In KPC tumours, the T-cell response was variable without a consistent change in infiltration following the combination of radiation and aCSF (Fig 4B).

Due to the low numbers of T cells in KPC tumours, we were only able to detail T-cell phenotypes in MC38 tumours. Radiation was associated with significantly elevated Ki67 expression, which did not increase with TAM depletion (Fig 4C). These data suggest that increased proliferation at least partly underlies the increase in T-cell numbers. Overall, the reduction in TAMs may contribute to a relative increase in CD8 numbers. However, despite the significant reduction in TAMs observed in KPC tumours CD8 numbers remained unchanged.

CD8 T cells exhibit features of exhaustion after extended exposure to target cells, limiting their cytotoxic potential (Yamamoto *et al*, 2008; Ahmadzadeh *et al*, 2009; Saito *et al*, 2013).

Programmed death 1 (PD-1) expression is one marker for exhaustion. Flow cytometry revealed small but significant decreases in PD-1 expression on T cells in both irradiation and combination therapy groups (Fig 4D). Finally, we analysed the effector status of T cells using IFN gamma as an activation marker and granzyme B as indicative of cytotoxic activity. Interferon gamma was significantly increased in T cells from irradiated tumours with macrophage depletion having little effect. Granzyme B positivity was only increased in irradiated tumours depleted of macrophages (Fig 4E and F).

The spatial distribution of T cells within tumours was assessed. In MC38 tumours, T cells were homogenously distributed throughout the tumours, and this pattern did not change with aCSF treatment (Fig 4G). In KPC tumours treated with irradiation alone, the few T cells identified were clustered in the tumour periphery (Fig 4H). In contrast, in the KPC tumours that received combination treatment, T cells were present throughout the tumour (Fig 4H, red boxes).

Depletion of CD8 T cells using a neutralising antibody in combination treatment groups completely abrogated the tumour growth delay observed in previous experiments (Fig 4I–K). Abrogation of the effect was also observed after experimental replication in immunodeficient mice, further confirming a T-cell-dependent effect (Fig 4L and M). These data substantiate the dependence of the increased tumour growth delay following TAM depletion on CD8 T cells. Furthermore, this phenomenon is associated with the spatial distribution as well as the number of CD8 T cells. Recently, administration of a CSF-1R inhibitor was reported to result in increased granulocytic MDSCs (Kumar *et al*, 2017). This was due to a loss of CSF-1-mediated suppression of chemokine secretion by fibroblasts. We did not identify significant changes in either neutrophil or gMDSC populations following aCSF (Appendix Figs S2A and C, and S3A and C).

**T-cell antigen priming is altered after irradiation**

Despite a significant increase in CD8 T cells infiltrating MC38 tumours following aCSF, there was no effect on tumour growth in the absence of irradiation. Therefore, we asked whether IR was involved in tumour-specific T-cell priming. Splenic CD8 T cells were

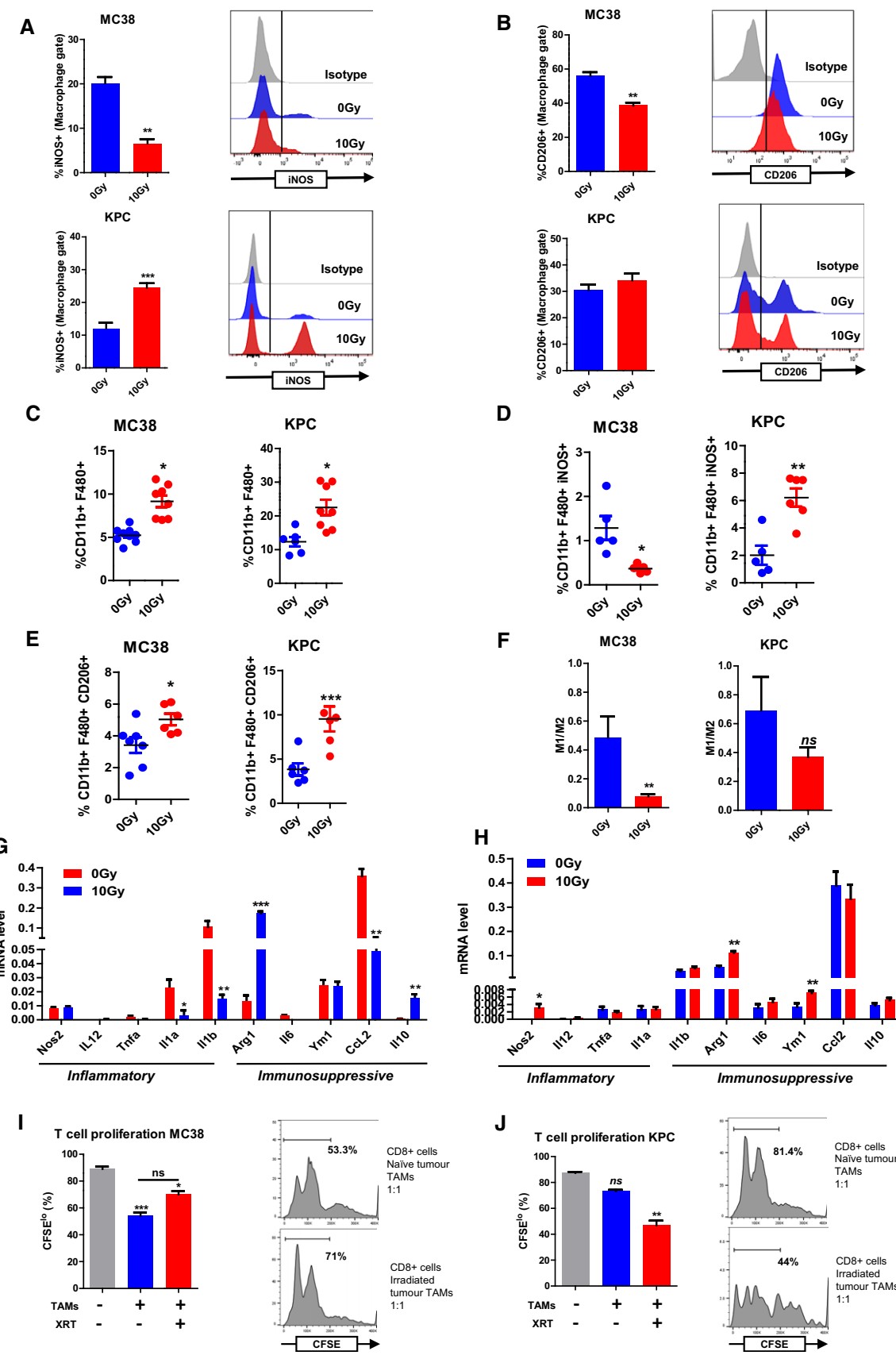

Figure 2.

**Figure 2. Macrophages recruited after irradiation are polarised towards an immunosuppressive, pro-tumourigenic phenotype.**

MC38 and KPC tumours were irradiated with 10 Gy and harvested after 5 days. Tumours were disaggregated, immune cells were identified by flow cytometry, and TAMs were isolated by FACS or magnetic bead separation.

A    Quantification of iNOS expression on gated macrophages (CD11b$^+$F480$^+$) from MC38 and KPC tumours, with representative histograms. Data are presented as mean ± SEM and analysed by Mann–Witney test ($n$ = 6 mice/group).

B    Quantification of CD206 expression on macrophages (CD11b$^+$F480$^+$) from MC38 and KPC tumours, with representative histograms. Data are presented as mean ± SEM and analysed by Mann–Witney test ($n$ = 6 mice/group).

C    Quantification of the percentage of TAMs (CD11b$^+$F480$^+$) in MC38 control tumours compared with irradiated tumours ($n$ = 6 mice/group). Data were analysed by Mann–Witney test.

D, E    Quantification of iNOS$^{hi}$ (D) and CD206$^{hi}$ (E) macrophages as a percentage of total cells in MC38 tumours ($n$ = 6). Data are presented as mean ± SEM and analysed by Mann–Witney test.

F    The total number of iNOS$^{hi}$ TAMs were divided by CD206$^{hi}$ TAMs to derive a M1/M2 ratio in MC38 and KPC tumours receiving mock or 10 Gy irradiation. Data are presented as mean ± SEM and analysed by Mann–Witney test ($n$ = 6 mice/group).

G, H    TAMs (CD11b$^+$F480$^+$) were isolated by FACS. Expression of selected immune stimulatory and immunosuppressive genes in TAMs was determined by RT–qPCR ($n$ = 3). Data are presented as mean fold change ± SEM compared to TAMs from mock-irradiated tumours ($n$ = 3). Statistical significance was determined by Mann–Witney test.

I, J    Assessment of TAM suppression of T cells was assayed by evaluation of inhibition of T-cell proliferation. TAMs were isolated by magnetic bead separation using F480 microbeads and co-cultured at a 1:1 ratio with CFSE-labelled CD8$^+$ T cells. CFSE dilution was analysed by flow cytometry to measure proliferation. Percentages of CFSE$^{lo}$ T cells were analysed by Kruskal–Wallis test with Dunn's multiple comparisons test. Representative histograms are shown (right panel). Experiments were repeated twice for each tumour cell line ($n$ = 3 mice/group, mean ± SEM).

Data information: *$P$ < 0.05, **$P$ < 0.01, ***$P$ < 0.001.
Source data are available online for this figure.

isolated from MC38 tumour-bearing mice and co-cultured with naïve or irradiated tumour cells in an ELISpot assay using interferon gamma as the readout. T cells from unirradiated tumour-bearing mice showed increased activity against irradiated tumour cells compared with control cells; however, this did not reach statistical significance (Fig 5A and B). T cells from mice bearing irradiated tumours showed a non-significant increase in activity against control tumour cells. The greatest increase in activity was in using T cells from mice bearing irradiated tumours tested against irradiated tumour cells. These results show that local tumour irradiation results in systemic T-cell priming. The primed T cell population recognized both irradiated and naïve tumour cells.

A key process in antigen-specific T-cell killing is the engagement of T-cell receptors (TCRs) with major histocompatibility complex I (MHCI) antigen complexes. MHCI expression was increased in MC38 cells after irradiation in culture (Fig 5C). Irradiation of KPC cells in culture resulted in induction of only a small population of MHCI-positive KPC cells (Fig 5D). In contrast, after irradiation of both MC38 and KPC tumours, MHCI expression increased (Fig 5E and F). In addition to MHCI, antigen-presenting cells (APCs) present antigens via MHC class II (MHCII) molecules. There was no decrease in MHCII$^+$ DCs following aCSF treatment (Fig 5G and H). MHCII$^+$ TAMs were significantly reduced following aCSF treatment (Fig EV4A and B). The CD8 T cells harvested from irradiated tumour groups with or without aCSF gave the same results in the ELISpot assay, indicating that the reduction in MHCII TAMs did not substantially affect antigen presentation (Fig EV4C and D).

To assess the systemic nature of the immune response after radiation and aCSF, we induced two tumours, one in each flank, allowing localised radiation treatment to only one tumour (Fig 5I) designated as the primary tumour. Tumours in the opposite flank were designated as the secondary tumour. In the MC38 model, when 10 Gy was applied to the primary tumour, secondary tumours continued to grow at a similar rate to unirradiated tumours (Fig 5J). The administration of aCSF to the mice resulted in a modest growth delay in the secondary, unirradiated tumours (Fig 5K). In KPC tumours, there was no significant growth delay observed in secondary tumours when primary tumours were treated with

irradiation alone (Fig 5L). In the combination treatment group, secondary tumours reached end-point by 8.75 days compared with 7 days for aCSF alone ($P$ = 0.03; Fig 5M). We examined the immune cell infiltrate present in the primary and secondary MC38 tumours by flow cytometry. Changes in macrophage and CD8 T-cell populations in the primary tumours were comparable to those observed in our previous experiments (Fig EV5A and B) in mice bearing only one tumour. However in secondary tumours, aCSF was less effective at reducing TAMs when the primary tumour received irradiation (Fig 5N). There was a trend towards increased CD8 T cells in the secondary tumours when the primary was treated with 10 Gy and the mouse received aCSF; however, this did not reach statistical significance. Additionally, the increase was less than that observed when mice bearing tumours were treated with aCSF alone (Fig 5O). The absence of a significant increase in CD8 T cells may be attributed to the relative decrease in sensitivity to aCSF observed in the secondary tumours. This phenomenon may also explain the more modest tumour growth delay observed in the secondary tumours.

These results are evidence of a modest, but significant abscopal effect. Whilst TAM depletion is associated with increased CD8 T-cell infiltration, it is the addition of irradiation which is key to an effective anti-tumour response.

## Macrophage depletion renders tumours sensitive to immune checkpoint blockade therapy

We now questioned possible limitations of the anti-tumour effects by immune checkpoint engagement. Radiation can induce PD-L1 expression on tumour cells, limiting a CD8-mediated anti-tumour response. Forty-eight hours following 10 Gy irradiation, PD-L1 was significantly increased on MC38 and KPC cells in tissue culture and *in vivo* (Fig 6A–D). At the same time, high levels of PD-L1 and PD-L2 were found on TAMs and were unaffected by irradiation (Fig 6E and F). MC38 tumours are known to be sensitive to PD-L1 blockade (Juneja *et al*, 2017; Lau *et al*, 2017). Here, combination treatment with IR and anti-PD-L1 resulted in complete tumour regression in 4/8 mice. The addition of aCSF did not increase the number of

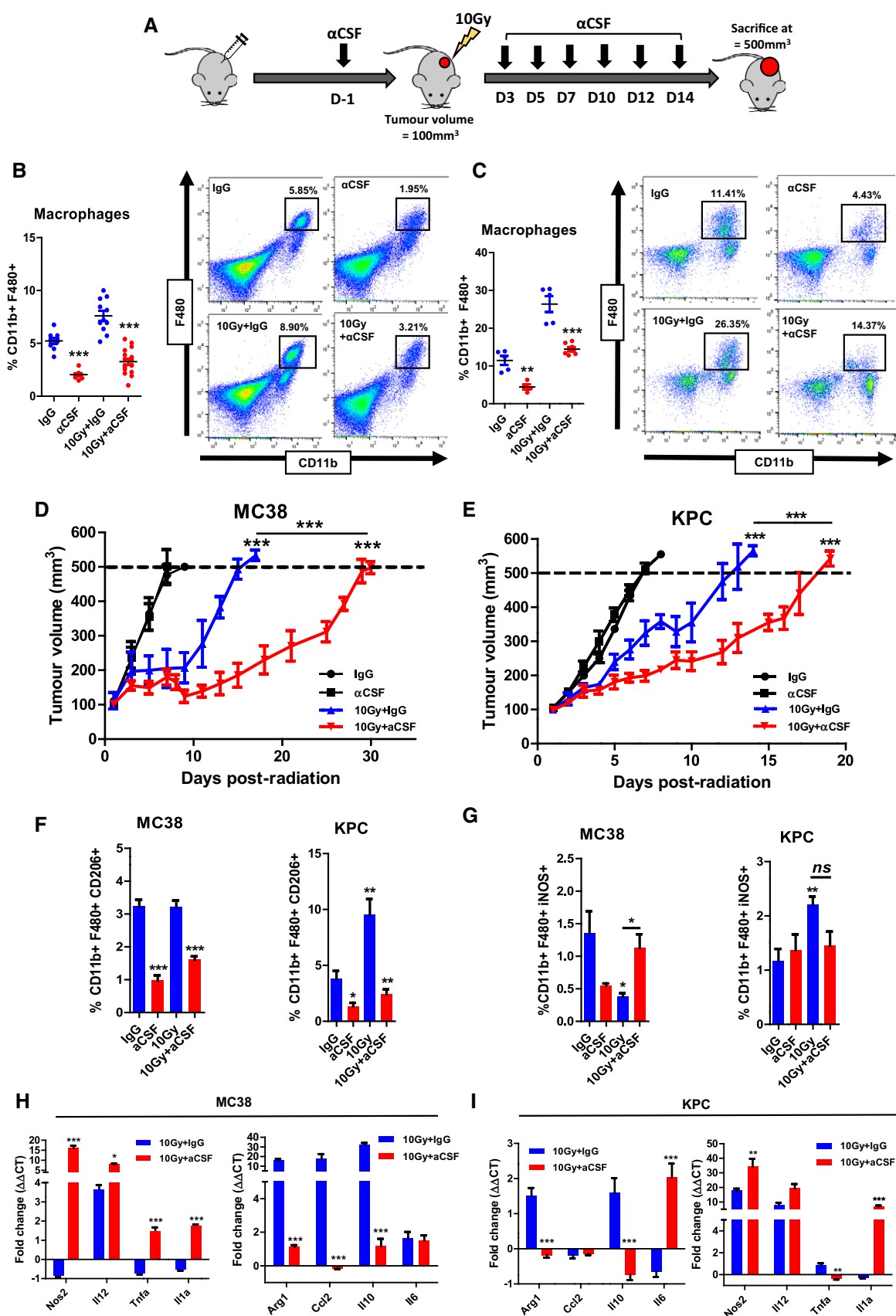

**Figure 3.**

◀

**Figure 3.  Anti-CSF therapy re-polarises macrophages and delays tumour growth following irradiation.**

A     The figure shows a schematic outlining the experimental approach. MC38 and KPC tumours were induced in the flank of C57BL/6 wild-type mice. When tumours reached 80 mm³, mice were randomly assigned to treatment groups and received antibody treatment (IgG or aCSF). When tumours reached 100 mm³, mice in the irradiation groups received 10 Gy to the tumours. For growth kinetics, a humane end-point was reached when tumours exceeded 500 mm³.

B, C   Flow cytometric analysis of TAMs (CD11b⁺F480⁺) in MC38 (B) and KPC (C) tumours receiving indicated treatments. Tumours were harvested 5 days following IR and disaggregated for analysis by flow cytometry. The left panels show the data derived from the flow cytometry with representative plots shown in the right panels. Data are presented as mean ± SEM and analysed by one-way ANOVA with Tukey's *post hoc* adjustment (*n* = 6 mice/group, three independent experiments).

D, E   Tumour growth kinetics of MC38 (D) and KPC (E) tumours receiving the indicated treatments. Data are presented as mean tumour volume ± SEM and analysed by one-way ANOVA with Tukey's *post hoc* adjustment (*n* = 6 mice/group).

F, G   Shows flow cytometric analysis of CD206ʰⁱ (F) and iNOSʰⁱ (G) TAMs in MC38 and KPC tumours 5 days following IR. Data are presented as mean ± SEM and analysed by one-way ANOVA with Tukey's *post hoc* adjustment (*n* = 6 mice/group).

H, I   CD11b⁺F480⁺ TAMs were isolated from MC38 (H) and KPC (I) tumours 5 days following irradiation (±aCSF), and expression of selected immune stimulatory and immunosuppressive genes was analysed by RT–qPCR (*n* = 3). Data are presented as mean ± SEM (10 Gy vs. 10 Gy + aCSF *n* = 6 mice/group). Statistical significance was determined by Mann–Witney test.

Data information: *$P < 0.05$, **$P < 0.01$, ***$P < 0.001$.
Source data are available online for this figure.

tumour regressions (Fig 6G). KPC tumours are highly resistant to immune checkpoint blockade, and we observed no tumour regression in mice treated with IR and anti-PD-L1 (Fig 6H; Winograd *et al*, 2015; Azad *et al*, 2016). However, the addition of aCSF led to tumour regression in three of eight tumours (Fig 6H). These results suggest that TAMs contribute to a hostile, immunosuppressive TME that potentiates resistance to immune checkpoint blockade. In order to determine whether tumour regression was dependent upon local tumour irradiation, we again utilised the double tumour model (Fig 5I).

There was no growth delay in contralateral tumours in the IR + anti-PD-L1 group (Fig 6I). However, in the triple combination group there was a small but significant increase in end-point (12 vs. 11 days $P < 0.05$, Fig 6J). Taken together, these data demonstrate that irradiation induces a highly suppressive tumour landscape due to increases in both tumour cell PD-L1 and PD-L1-rich TAMs. Combination therapy may be deployed in situations where immune checkpoint blockade is currently ineffective.

## Discussion

Radiation of tumours stimulates anti-tumour immunity, yet often fails to generate effective anti-tumour responses. In the present study, we show that the recruitment of macrophages after radiation of tumours is one component resisting the induction of immunity. Depletion of these macrophages using aCSF significantly delays tumour regrowth following radiation due to enhanced adaptive immunity. Growth inhibition was constrained further by radiation-induced upregulation of PD-L1 on cancer cells, coincident with concurrent high PD-L1 expression on macrophages so that addition of anti-PD-L1 blocking antibody to aCSF treatments extended the growth delay induced by radiation with regression in a subset of tumours. Radiation had a stimulatory effect on anti-tumour immunity through augmentation of antigen-specific T-cell priming. Together, these data demonstrate that radiation has the capacity to elicit an adaptive immune response balanced by the induction of immunosuppressive macrophages limiting effective tumour eradication.

Colony-stimulating factor 1 was induced by radiation of the cancer cells and their tumours. CSF-1 acting through its receptor CSF-1R is essential for the differentiation, recruitment and ultimately survival of macrophages derived from immature monocytes. Many factors contribute to CSF-1 expression (Harrington *et al*, 1997; Song *et al*, 2007; Chen *et al*, 2008; Wittrant *et al*, 2008). In the context of tumour irradiation, Xu *et al* (2013) reported that ABL1 was translocated to the nucleus, binding to the CSF-1 promoter region resulting in increased transcription of CSF-1. The transient induction of tumour cell CSF-1 gene expression was reflected in a similar pattern of protein secretion *in vivo*, which may be explained by the short period of cell viability following radiation before mitotic catastrophe or apoptosis results in tumour cell death. Critically, the dependence of macrophages on CSF-1 for survival makes CSF-1(R) blocking agents attractive candidates for use in the clinical setting and there are already numerous actively recruiting clinical trials (Ries *et al*, 2014).

In the literature, the effect of radiation on both the recruitment and functional status of macrophages appears to be dependent on the experimental model, radiation dose and the time at which tumours are analysed. Whilst some reports find recruitment of macrophages (Kozin *et al*, 2010; Crittenden *et al*, 2012; Xu *et al*, 2013), others do not identify any significant change (Zaleska *et al*, 2011; Deng *et al*, 2014b). In general, macrophages are increased after irradiation in murine tumours as early as 24 h, peaking after 1–2 weeks and slowly decreasing to baseline levels (Crittenden *et al*, 2012; Shiao *et al*, 2015; Seifert *et al*, 2016). We found considerable increases in macrophages within days following radiation, coinciding with increased CSF-1. The reduction in macrophages over time suggests a diminution of the initial stimulus responsible for recruitment. In addition to recruitment, radiation can affect the gene expression and function of macrophages. Shiao *et al* (2015) analysed tumour macrophages harvested 24 h following 5 Gy irradiation finding upregulation of genes in both pro-inflammatory and immunosuppressive pathways, suggestive of generalised activation. Murine (KC) pancreatic tumours from genetically engineered models and allografts showed a significant shift towards M2 polarisation following radiation (Crittenden *et al*, 2012; Seifert *et al*, 2016). Our results highlight the heterogeneous nature of response between tumour types, with a more inflammatory phenotype in KPC tumours compared to MC38, though the general trend is towards M2, and here, in both cases aCSF led to enhanced anti-tumour immunity.

In our hands, treatment of mice with aCSF reduced TAMs by approximately half. Whether aCSF itself is only partially effective,

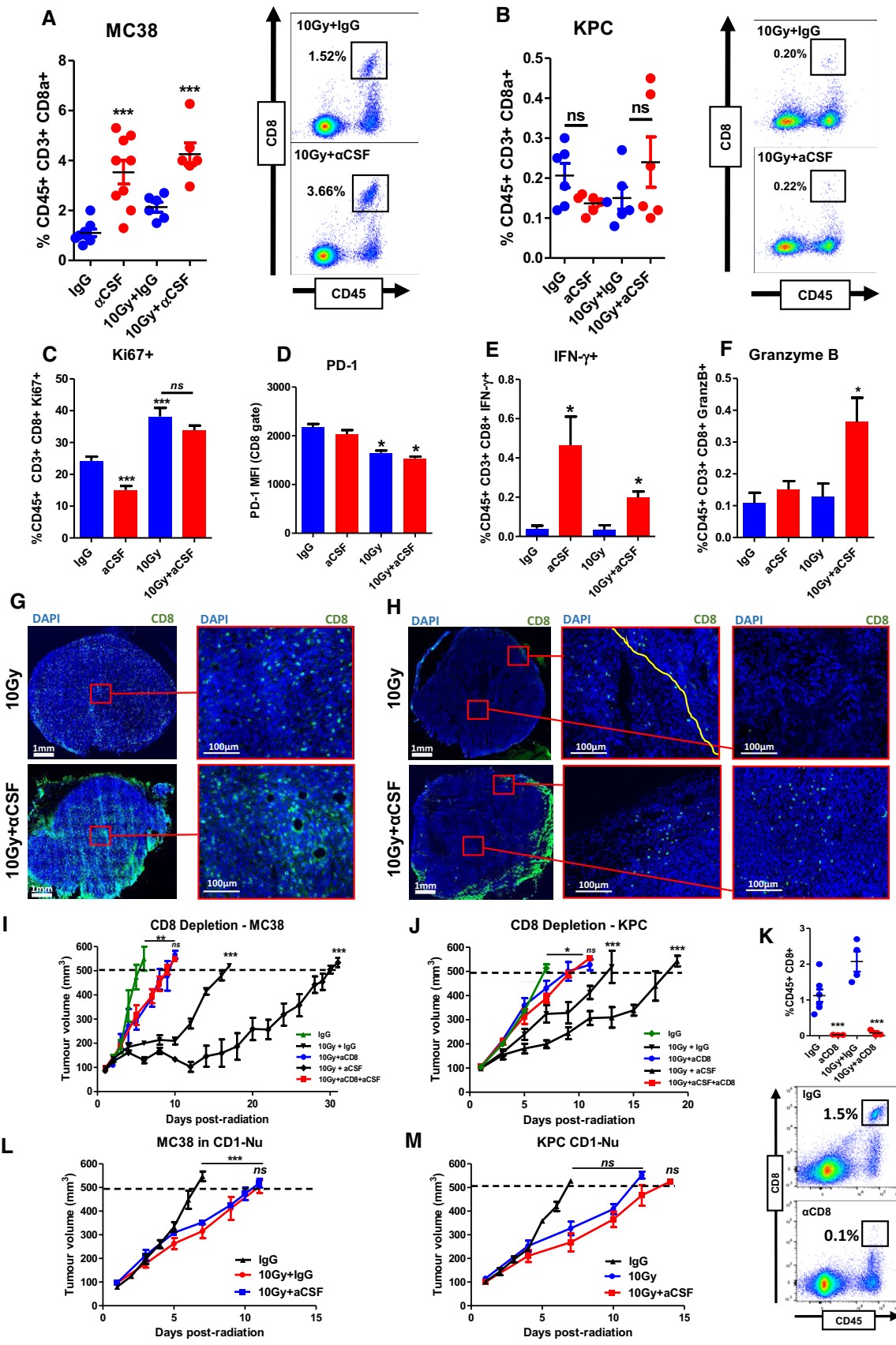

**Figure 4.**

**Figure 4. Cytotoxic CD8 T lymphocytes infiltrate macrophage-depleted tumours.**

A, B    MC38 (A) and KPC (B) tumours were harvested 5 days following 10 Gy IR ± aCSF as indicated. The left panels show the percentage of CD45$^+$CD3$^+$CD8$^+$ T cells in these tumours after the indicated treatments. Representative flow cytometry plots from the irradiated groups are shown in the right panels. Tumours that did not receive irradiation were harvested when tumours reached 500 mm$^3$. Data are presented as mean ± SEM and analysed by one-way ANOVA with Tukey's *post hoc* adjustment (A) and Kruskal–Wallis test with Dunn's multiple comparisons test (B) (*n* = 6 mice/group, three independent experiments).

C    Flow cytometric analysis of Ki67 expression in the CD8$^+$ T cells in MC38 tumours from A. Data are presented as mean ± SEM and analysed by Kruskal–Wallis test with Dunn's multiple comparisons test (*n* = 6/group).

D    Flow cytometric analysis of PD-1 expression on CD8$^+$ T cells from MC38 in tumours as in (A). Data are presented as mean ± SEM and analysed by Kruskal–Wallis test with Dunn's multiple comparisons test (*n* = 6/group).

E, F    Flow cytometric analysis of IFN-γ and granzyme B expression in CD8$^+$ T cells isolated from MC38 tumours in (A). Data are presented as mean ± SEM and analysed by Kruskal–Wallis test with Dunn's multiple comparisons test (*n* = 6 mice/group, three independent experiments).

G, H    Immunofluorescent staining of MC38 (G) and KPC (H) tumour sections, blue = DAPI, green = CD8. Yellow line demarcates the tumour capsule.

I, J    Tumour growth in CD8-depleted C57BL/6 wild-type mice bearing MC38 (I) and KPC (J) tumours receiving treatment as indicated (*n* = 6 mice/group). Data are presented as mean ± SEM and analysed by one-way ANOVA with Tukey's *post hoc* adjustment.

K    Flow cytometric quantification of intra-tumour CD8 T cells following anti-CD8 treatment. Data are presented as mean ± SEM and analysed by unpaired *t*-test (*n* = 5 mice/group).

L, M    Tumour growth in athymic nude mice bearing MC38 (I) and KPC (J) tumours receiving treatments as indicated (*n* = 6 mice/group). Data are presented as mean tumour volume ± SEM and analysed by one-way ANOVA with Tukey's *post hoc* adjustment.

Data information: *$P$ < 0.05, **$P$ < 0.01, ***$P$ < 0.001.
Source data are available online for this figure.

whether there are redundant mechanisms of recruitment or whether a subset of macrophages are resistant to CSF-1 depletion remains to be determined. In our experiments, the refractory population of macrophages were polarised towards an inflammatory state, resulting in an increased "M1:M2" ratio. These macrophages may be more resistant due to reduced CSF-1R expression, or reflect a population that has not yet been polarised by the tumour microenvironment. Similar findings have been reported following the application of CSF-1 blockade, with a consistent pattern of significantly reduced arginase expression (Pyonteck *et al*, 2012; Zhu *et al*, 2014; Shiao *et al*, 2015; Seifert *et al*, 2016).

Arginase (Arg-1) is a well-defined M2 marker. Arg-1 was present at high baseline levels in TAMs and BMDMs co-cultured with tumour cells, suggesting that the tumour cells themselves help condition the macrophages towards an immunosuppressive phenotype. Arg-1 in tumour macrophages or co-cultured macrophages further increased following irradiation of the tumour or radiation of the tumour cells used in co-culture, respectively. Arginase-mediated L-arginine depletion can profoundly limit T-cell function and metabolism (Shiao *et al*, 2015; Seifert *et al*, 2016) (Geiger *et al*, 2016), which may underlie our finding of enhanced macrophage-mediated T-cell suppression following radiation. In the context of existing reports, it appears that whilst some transient alterations in inflammatory gene expression appear early in the radiation response, the overwhelming effect is a significant increase in predominantly immunosuppressive macrophages.

The immunosuppressive function of the infiltrating macrophages was revealed by their depletion. aCSF does not directly target T cells, yet depletion of macrophages led to significant increases in T-cell infiltration. In aCSF-treated mice bearing MC38 tumours, there was a twofold increase in CD8 T cells. Consistent with other reports, we found very few CD8 T cells in KPC tumours (~0.15%) and no detectable increase following aCSF. Tumour penetration was evident in the central region of the tumours where T cells were absent in untreated KPC tumours. The presence of T cells at the tumour core compared with tumour margins is associated with improved outcomes (Galon *et al*, 2006; Berthel *et al*, 2017; Chen & Mellman, 2017). Others have also reported the surprising ability of

**Figure 5. T-cell antigen priming is enhanced by irradiation.**

A, B    CD8$^+$ T cells were isolated from the spleens of MC38 tumour-bearing mice. The tumours were radiated with 10 Gy, and cells were harvested 5 days later. Quantification (A) and representative images (B) of MC38 tumour cell-specific tumour-derived CD8 T-cell responses detected by IFN-γ ELISpot. The tumour-specific CD8$^+$ T-cell response was evaluated after T-cell incubation with naïve or irradiated MC38 cells for 24 h. Data are presented as mean ± SEM and analysed by Kruskal–Wallis test with Dunn's multiple comparisons test (*n* = 3 mice/group).

C, D    Flow cytometric detection of major histocompatibility complex I (MHCI) expressed on MC38 (C) and KPC (D) tumour cells 48 h following exposure to 10 Gy IR. The left graph shows the overall data, with representative flow cytometry plots on the right. Data are presented as mean ± SEM and analysed by Mann–Witney test (*n* = 3/group).

E, F    Flow cytometric quantification of major histocompatibility complex I (MHCI) expression *in vivo*. Gated MC38 (E) and KPC (F) tumour cells were analysed 48 h following exposure to 10 Gy IR. Data are presented as mean ± SEM and analysed by unpaired *t*-test (*n* = 3/group).

G, H    Flow cytometric quantification of dendritic cells (CD11b$^+$CD11c$^+$MHCII$^+$) in MC38 (G) and KPC (H) tumours receiving treatment as indicated and as in (E, F). Data are presented as mean ± SEM and analysed by unpaired *t*-test (*n* = 5 mice/group).

I    Schema outlining double tumour model (see Materials and Methods).

J, K    Tumour growth in mice bearing two MC38 tumours receiving 10 Gy IR to the primary lesion (J) ± systemic aCSF therapy (K). The differences in tumour volume 9 days following IR are presented as mean ± SEM and analysed by unpaired *t*-test (*n* = 5 mice/group).

L, M    Tumour growth in mice bearing two KPC tumours receiving 10 Gy IR to the primary lesion (J) ± systemic aCSF therapy (M). The difference in mean tumour volume 10 days following IR are presented as mean ± SEM and analysed by unpaired *t*-test (*n* = 8 mice/group).

N, O    Flow cytometric analysis of macrophages (N) and CD8 T cells (O) in primary and secondary MC38 tumours. Data are presented as mean ± SEM and analysed by Kruskal–Wallis test with Dunn's multiple comparisons test (*n* = 5 mice/group).

Data information: *$P$ < 0.05, **$P$ < 0.01, ***$P$ < 0.001.
Source data are available online for this figure.

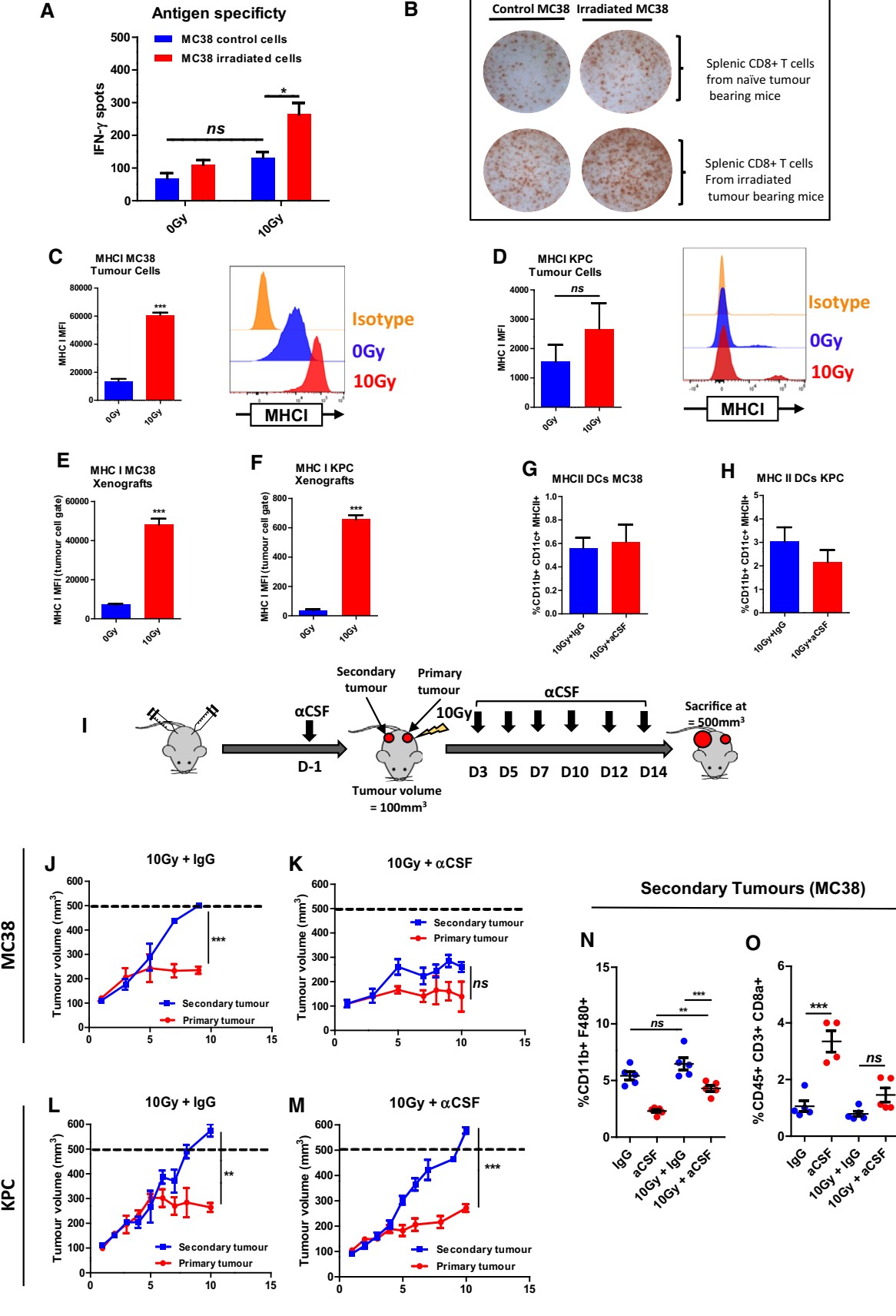

Figure 5.

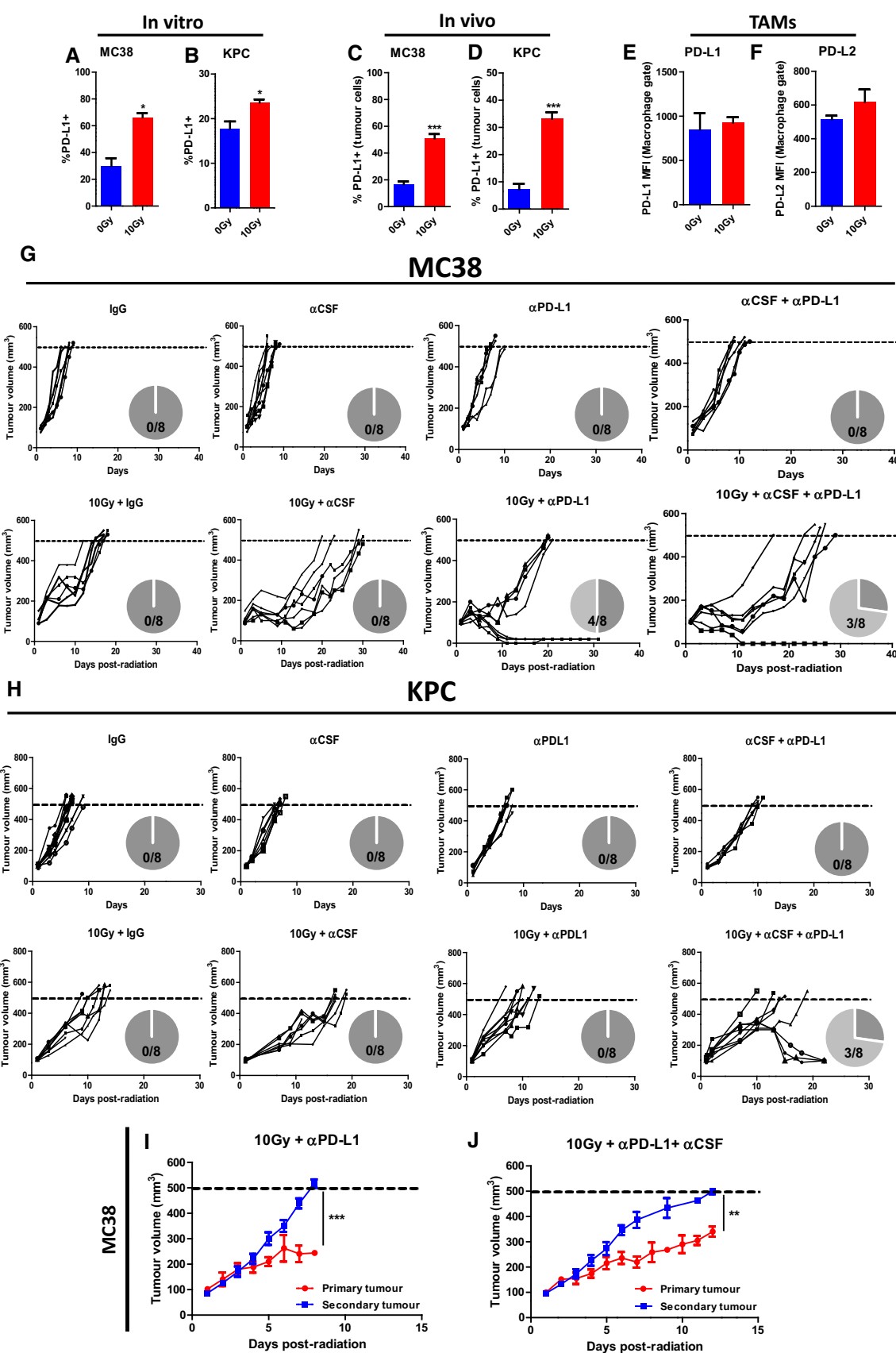

**Figure 6.**

◄

**Figure 6. Macrophage depletion renders tumours more responsive to immune checkpoint blockade therapy.**

A–D  PD-L1 expression on MC38 and KPC cells 48 h following 10 Gy irradiation in tissue culture (A, B) or 10 Gy irradiation of tumours (C, D) analysed by flow cytometry. Data are presented as mean ± SEM and analysed by Mann–Witney test (n = 3, A, B). Data are presented as mean ± SEM and analysed by unpaired *t*-test (n = 5 mice/group, C, D).

E, F  Flow cytometric analysis of PD-L1 (E) and PD-L2 (F) on TAMs in MC38 tumours receiving treatment as indicated above. Data are presented as mean ± SEM and analysed by Mann–Witney test (n = 5 mice/group).

G, H  Tumour growth in mice bearing MC38 (G) and KPC (H) tumours receiving the indicated treatments. Data presented for individual mice. Pie charts indicate the number of regressions observed.

I, J  Tumour growth in mice bearing KPC tumours mice receiving 10 Gy IR + systemic aPD-L1 to the primary lesion (I) ± systemic aPD-L1 + aCSF therapy (J). The difference in tumour volume 8 (I) or 10 (J) days following IR was analysed by unpaired *t*-test (data presented as mean ± SEM, n = 8 mice/group).

Data information: *$P < 0.05$, **$P < 0.01$, ***$P < 0.001$.

very few T cells to mount a potent immune response in KPC tumours (Evans *et al*, 2016). In general, increased T-cell numbers follow CSF-1(R) blockade in a variety of tumour models, but rarely results in growth inhibition without additional therapies (Strachan *et al*, 2013; Mok *et al*, 2014; Zhu *et al*, 2014; Holmgaard *et al*, 2016; Seifert *et al*, 2016). For example, Zhu *et al* (2014) found that combining CSF-1R blockade with anti-CTLA4 or PD-L1 resulted in significant growth inhibition in orthotopic pancreatic tumours. Holmgaard *et al* (2016) used the same agent in combination with indoleamine 2,3-dioxygenase (IDO) inhibitors, and Mok *et al* (2014) found that CSF1R blockade significantly improved CD8 T-cell infiltration and activity following adoptive T-cell therapy. There is consensus amongst these reports that greater T-cell activity was due to a reduction in suppressive macrophages; however, the exact mechanism remains unclear. Strikingly, despite increased T-cell infiltration resulting from aCSF alone, we did not observe anti-tumour activity unless aCSF was combined with radiation.

We examined the possibility that radiation improved T-cell priming accounting for its effect on immunity after aCSF treatment. This concept emerged following clinical reports of anti-tumour effect outside of the radiation field, the so called "abscopal effect". Since then, a number of studies have demonstrated radiation-dependent T-cell priming, though often using exogenous tumour peptides such as ovalbumin (Lugade *et al*, 2005; Lee *et al*, 2009; Schaue *et al*, 2012; Sharabi *et al*, 2015). More recently, Rudqvist *et al* (2018) show a radiation-dependent increase in the number and diversity of T-cell receptor clones. We found that splenic CD8 T cells isolated from mice bearing irradiated tumours were significantly more active towards irradiated tumour cells compared with naïve cells *in vitro*, suggesting increased presentation of peptides but not excluding additional effects of increased DAMPs. Interestingly, in mice bearing bilateral tumours, irradiation alone did not result in growth inhibition in the unirradiated tumour. These data suggest that whilst radiation alone is able to augment antigen-specific priming, this is not sufficient. Addition of systemic aCSF therapy can improve local infiltration and activity of T cells.

In the absence of tumour regression, we questioned whether a T-cell response was additionally limited by the engagement of immune checkpoint, potentially exacerbated by the upregulation of checkpoint molecules following radiation (Deng *et al*, 2014a; Azad *et al*, 2016; Derer *et al*, 2016). In our models, both PD-L1 and PD-L2 were already expressed at high levels on macrophages regardless of radiation. PD-L1 expression on tumour cells was increased by radiation. Nonetheless, the addition of anti-PD-L1 did not improve the response in MC38 tumours, but interestingly, further growth inhibition and in some cases regression were observed in KPC tumours. MC38 is microsatellite unstable, hypermutated, immunogenic and

has shown sensitivity to immune checkpoint blockade (Deng *et al*, 2014a; Juneja *et al*, 2017; Lau *et al*, 2017). Conversely, KPC tumours fail to generate robust adaptive immunity and are highly resistant to checkpoint blockade (Azad *et al*, 2016; Evans *et al*, 2016). In addition, the relative contribution of host vs. tumour cell expression of PD-L1 to the sensitivity of tumours is different across different tumour types (Juneja *et al*, 2017; Lau *et al*, 2017). These data, together with our observation of significantly more macrophages in the KPC model, may explain the advantage of triple therapy.

In summary, we show that adaptive immunity induced by radiation is limited by the recruitment of highly M2-polarised immunosuppressive macrophages. Macrophage depletion partly reduced the immunosuppression after radiation, but additional treatment with anti-PD-L1 was required to achieve tumour regression. Even with both aCSF and aPD-L1 treatment and radiation however, some mice failed to generate effective anti-tumour responses. This work demonstrates that radiation-induced immunity is limited by a suppressive microenvironment. The immunosuppressive response can be partially abrogated by aCSF-mediated alteration in macrophage infiltration and by PD-L1 checkpoint inhibition.

# Materials and Methods

### Tumour challenge and treatment experiments

Animal procedures were in accordance with UK Animal law (Scientific Procedures Act 1986), including local ethics approval. Female, C57BL/6 wild-type (6–8 weeks) and CD1-nude (8–10 weeks) mice were purchased from Charles River laboratories (Kent, UK) and housed in a pathogen-free facility with 12-h light cycles. KPC cells were derived from KrasLSL$^{G12D}$/+;p53$^{R172H}$/+;Pdx1-Cretg/+ (KPC) tumours. MC38 cells were purchased from American Type Tissue Collection (ATCC). Cell line authentication was performed using Short Tandem Repeat profiling (Cancer Research UK genomic facility, Leeds Institute of Molecular Medicine, March 2014). All cell lines were negative for mycoplasma (Lonza Mycoalert™ Test kit). MC38 ($0.5 \times 10^6$) or KPC ($0.25 \times 10^6$) cells were injected into the flank(s) of anaesthetised mice. Tumours were measured daily in three dimensions using digital callipers, and volume was calculated using the formula $0.5 \times$ Length $\times$ Width $\times$ Height. When tumours reached 80 mm³, mice were randomly assigned to treatment groups. Anti-CSF (Bioxcell, clone 5A1) was administered intraperitoneally at a dose of 10 mg/kg three times weekly, anti-PD-L1 (Bioxcell, clone 10F.9G2) at 10 mg/kg on days −1, 3, 6 and 9 and anti-CD8a (Bioxcell, clone 2.43) at 250 µg on days −1, 3, 6 and 9. Radiation was initiated when tumours reached 100 mm³, delivered via a Gulmay 320 irradiator.

**The paper explained**

**Problem**
Radiation can both stimulate and suppress immunity. The stimulatory effects of radiation offer the potential for it to augment novel anti-cancer therapies. However, the immunosuppressive effects first need to be thwarted in order for these benefits to be unleashed.

**Results**
We show that radiation stimulated the release of colony-stimulating factor 1 (CSF-1) by tumour cells. Increased CSF-1 was associated with increased tumour-associated macrophages (TAMs), which were immunosuppressive. TAMs were effectively depleted by the administration of anti-CSF antibody. Remaining TAMs were repolarised to an immune stimulatory phenotype. These changes were associated with increased and more cytotoxic CD8$^+$ T cells. In pancreatic tumours (KPC) resistant to immune checkpoint blockade, triple combination therapy (10 Gy IR, aCSF and aPD-L1) led to regression of many tumours.

**Impact**
Resistance to immune checkpoint blockade has resulted in increased interest in combination therapies. Combining checkpoint blockade with radiotherapy has been shown to improve responses in some tumours. Our results emphasise the importance of accounting for microenvironmental alterations that take place after irradiation. Targeting specific inhibitory populations, in this case TAMs, demonstrates that rationalised combination therapy could be clinically useful in selected settings.

## Immunofluorescent staining

Sections were fixed in ice-cold acetone, rehydrated, and blocked with 20% goat serum, and primary antibody was incubated for 2 h at room temperature. Antibodies were directed against CD8 (Abcam, 22378) and CD68 (GeneTex, GTX41865). Secondary antibody staining was performed with Alexa Fluor 546 (Life Technologies). Sections were mounted using the ProLong® Diamond Antifade Mountant with DAPI (P36962; Fisher). Immunofluorescence was visualised utilising an inverted epifluorescence microscope (DM IRBE, Leica Microsystems).

## ELISA and cytokine arrays

Colony-stimulating factor 1 levels were determined with a mouse ELISA kit (MBS701429, MyBioSource), which was used according to the manufacturer's instructions. A proteome profiler array panel A (R+D systems) was used to analyse tumour cell conditioned media.

## Flow cytometry and flow-assisted cell sorting

Tumours were manually dissociated, incubated in Hank's balanced salt solution with 200 μl Collagenase II (Worthington, UK) on a shaker at 37°C for 40 min and passed through a 70-μm filter. After blocking with FcγIII/II (aCD16/32), surface antigen staining was performed. For intracellular staining, the eBiosciences FOXP3 intracellular staining kit was used according to the manufacturer's instructions (00-5523-00). For T-cell stimulation, $1 \times 10^6$/ml of cells were incubated in RPMI with 10% foetal bovine serum with

2 μl/ml of Cell Stimulation Cocktail with Brefeldin A (Biolegend, 423304) for 4 h. Antibodies used are listed in Appendix Table S1. Data were acquired on a BD FACSCanto™ II or Thermo Fisher Attune® NxT. Data were analysed using FlowJo, version 10.0. Gating strategies for immune cell populations can be seen in Appendix Fig S1. Cells were sorted using the Beckman Coulter Legacy MoFlo MLS Cell Sorter.

## Real-time quantitative PCR

RNA was extracted from samples using TRIzol according to the manufacturer's guidance. For FACS samples, PicoPure RNA isolation kit (Thermo Fisher) was used according to the manufacturer's instructions. TURBO DNA-free kit was used to eliminate genomic DNA (Thermo Fisher). RNA samples were reverse-transcribed using Tetro high capacity RNA to cDNA synthesis kit according to the manufacturer's protocol (Thermo Fisher). For each replicate, 25 ng of cDNA was loaded with SYBR Green (Thermo Fisher) and amplified in the following conditions: 40 cycles at 95°C (15 s), 60°C for 30 s and 72°C for 30 s. mRNA expression and fold change were analysed using the delta ct method, normalising for the housekeeping gene (β-actin). Primer pairs are listed in Appendix Table S2.

## Bone marrow-derived macrophage (BMDM) culture

Bone marrow was harvested from the femurs of C57BL/6 wild-type mice under sterile conditions. $3 \times 10^6$ cells were re-suspended in RPMI supplemented with penicillin (100 μg/ml) and streptomycin (100 μg/ml), 10% foetal bovine serum and 20% L929 conditioned medium and incubated for 5 days. For co-culture experiments, differentiated macrophages were seeded into 6-well plates ($1 \times 10^6$/ well). Tumour cells were seeded into Millicell® 0.4-μm cell culture inserts ($0.5–1.0 \times 10^6$/well) 24 h prior to transfer to the 6-well plates.

## T-cell suppression assay

CD8$^+$ T cells were isolated from the spleens of C57BL/6 wild-type mice using magnetic bead separation according to the manufacturer's instructions (Miltenyi Biotec). T cells were labelled with Cell-Trace CFSE cell tracking dye before being seeded into 96-well plates coated with anti-CD3/anti-CD28. RPMI was supplemented with L-glutamine, β-mercaptoethanol and recombinant interleukin-2. Tumour-derived macrophages were added at ratios of 1:1 and cultured overnight. CFSE signal in T cells was analysed by flow cytometry.

## IFN-γ ELISpot assay

The ebiosciences interferon gamma ELISpot kit was used according to the manufacturer's instructions. Spleens were harvested from tumour-bearing mice, control, aCSF-, 10 Gy- and 10 Gy+aCSF-treated groups. CD8 T cells were isolated and seeded in a sterile 96-well high protein binding Immobilon-P membrane culture plate (Millipore) with control or irradiated (10 Gy) MC38 tumour cells ($1 \times 10^5$ cells at a ratio of 1:1). Recombinant mouse

*Keaton I Jones et al*     Macrophages limit radiation immunity

**EMBO Molecular Medicine**

interferon gamma (Biolegend, UK) was used as a positive control. IFN-γ spots were quantified using the ELISpot plate reader (Oxford Biosystems).

### Statistical analysis

GraphPad Prism 5 was used for all data analysis. Unless otherwise indicated, data are presented as mean ± standard error (SEM). Statistical significance was determined if $P < 0.05$. Exact $P$-values are provided in Appendix Table S3. The statistical test used in individual experiments is indicated in the figure legends. To determine Gaussian distribution, data were analysed by the D'Agostino-Pearson Omnibus normality test. For parametric data, two-tailed unpaired Student's $t$-test (two groups) and one-way ANOVA with Tukey's *post hoc* adjustment (> 2 groups) were used. For non-parametric data, Mann–Whitney (two groups) and the Kruskal–Wallis (> 2 groups) tests with Dunn's multiple comparisons test were used. In animal experiments, all mice were randomly assigned to treatment groups. All animal experiments were conducted a minimum of twice, with $n$ referring to the number of biological replicates.

**Expanded View** for this article is available online.

### Acknowledgements
We thank Dr Andrew Worth, Jenner Institute, University of Oxford, for their help and dedication in the flow-assisted cell sorting experiments. This study was supported by Cancer Research UK (grant number C5255/A15935). Keaton Jones was supported by a CRUK Clinical Research Training Fellowship.

### Author contributions
RJM, KIJ and ANG-W conceived the study. KIJ, JT, JI, AY, ANG-W and JB performed experiments, and collected and analysed data. KIJ and RJM wrote the manuscript. All authors reviewed the manuscript.

### Conflict of interest
The authors declare that they have no conflict of interest.

### For more information
(i) https://www.ncbi.nlm.nih.gov/pubmed/26598942
(ii) https://www.ncbi.nlm.nih.gov/pubmed/28159861

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
