## [Review Process File · EMBO Molecular Medicine]

Radiation Combined with Macrophage Depletion Promotes Adaptive Immunity and Potentiates Checkpoint Blockade

K.I. Jones, J. Tiersma, A.E. Yuzhalin, A.N. Gordon-Weeks, J. Buzzelli, J. I'm, R.J. Muschel

Review timeline:

Submission date:	18 May 2018
Editorial Decision:	18 July 2018
Revision received:	12 September 2018
Editorial Decision:	25 September 2018
Revision received:	17 October 2018
Accepted:	19 October 2018

Editor: Céline Carret

Transaction Report:

1st Editorial Decision

18 July 2018

Thank you for the submission of your manuscript to EMBO Molecular Medicine and for your patience while your article was being reviewed. We have now heard back from the two referees whom we asked to evaluate your manuscript.

You will see from the comments pasted below that the referees found the study to be of potential interest. However, they both highlight a number of limitations that currently preclude further processing of your article. Many controls are either missing or unclear, figures and legends are inappropriate or confusing, which along with inadequate statistics and lack of essential details, result in a paper that is not sufficiently convincing. This said, given the importance of combination therapies and the encouraging results regarding tumor growth, we would welcome the submission of a revised version within three months for further consideration and would like to encourage you to address all the criticisms raised as suggested to improve conclusiveness and clarity.

Please note that EMBO Molecular Medicine strongly supports a single round of revision and that, as acceptance or rejection of the manuscript will depend on another round of review, your responses should be as complete as possible.

I look forward to receiving your revised manuscript.

***** Reviewer's comments *****

Referee #1 (Remarks for Author):

Combination therapies involving radiotherapy and checkpoint inhibitors, or CFS-1/CFS-1R have demonstrated activity in multiple tumor types, in clinical and pre-clinical studies, therefore although the triple combination has not been explored yet, the study is not totally innovative.

In the first paragraph of "Results", describing the results of FACS analysis the authors state that the number of macrophages, as well as of other immune subsets increases following irradiation. Since FACS analysis gives relative and not absolute results the word number should be changed in percentages and the authors should more carefully comment the results as increased infiltration. There is also another possibility i.e. that irradiation kills more tumor cells than immune cells, and that especially macrophages, which are not highly proliferating cells, are more spared than tumor cells and their percentage is relatively increased.

Legend to Fig. 1 has some mistakes in the indications of panels E, F, G, I. In the legend of panel F it is written that "Dashed lines reflect mean TAMs within control tumors", while in the figure there are no dashed lines.

Fig. 2. In Fig. 2A While in KPC cells the histogram plot show two clear peaks representing iNOS positive and negative cells, there is only one peak in MC38 tumors (corresponding to isotype), showing that there are no iNOS expressing macrophages neither in the control nor after irradiation. I therefore cannot understand where the percentages of iNOS positive macrophages shown in the graph come from. Starting from this observation, strong doubts arise about the reliability of the graphs shown in Fig. 2D and 2F (ratios of M1/M2 macrophages).

There are also some concerns regarding the comment to Fig. 2, the authors state that irradiation leads to increased T cell suppression (Figure 2 I,J). Nevertheless, in the case of T cells incubated with MC38-derived TAM after irradiation, the cell proliferation is significantly increased with respect to non-irradiated MC38-derived TAM.

Fig. 4. The authors' statement that increased T cell numbers are observed after anti-CSF treatment should be rephrased and contemplate the possibility that the percentage of T cells is relatively increased with respect to macrophages which are indeed diminished by the treatment.

In the tumor growth experiments shown in Fig. 4I 4J 4L 4M, it would be advisable to show also the growth of untreated tumors.

Fig. 6. In the tumor growth experiments one control is lacking. The authors should show the tumor growth after treatment with anti-PD-L1 and anti-CSF. It appears to me that in MC38 tumors there is no additive antitumor effect in adding a-PD-L1 to the combination of irradiation and anti-CSF. The appropriate control is needed to understand whether the triple combination is necessary or if anti-PD-L1 and anti-CSF are effective in double combination.

The statistical analysis is not convincing throughout the whole manuscript. In fact, most results have been analyzed by a parametric test (i.e. T-test) but there is no mention of the analysis of the normal distribution of data. A non-parametric test should be used.

Referee #2 (Remarks for Author):

In this manuscript, the authors investigated the function of macrophages in tumors treated by irradiation. They found that there was increased CSF1 secretion by tumor cells upon irradiation and there was also a concurrent increase of macrophages in irradiated tumors which displayed an immunosuppressive phenotype. When combined with radiation, macrophage depletion by anti-CSF1 delayed tumor growth compared to animals received radiation only. And this delay was abolished by depleting CD8 T cells via anti-CD8. Moreover, radiation induced higher PD-L1 expression on tumor cells. Combined treatment with irradiation + systemic aPD-L1 + aCSF1 led to tumor regression in 3 out of 8 KPC tumors, which are highly resistant to irradiation or aPD-L1 treatment alone.

Overall, the authors showed that irradiation leads to a highly immunosuppressive tumor microenvironment by recruiting tumor-associated macrophages and inducing PD-L1 expression in tumor cells. Importantly, combined therapy showed promising tumor regression phenotype in mouse KPC tumors. The authors need to address the following issues:

Main comments:

1. The authors stated that "CD8 completely abrogated the tumor growth delay observed in previous experiments" but data shown in Fig. 4I, J, L, M was not sufficient to show the abolition. The authors should include tumor growth curves of animals treated with (1) IgG only, (2) 10Gy+IgG and (3) 10Gy+aCSF.
2. Given that there was little CD8 T cell infiltration in KPC as shown in Fig. 4B and H. Could the authors comment on why the delayed tumor growth in aCSF treated KPC mice was abrogated by aCD8 treatment (Fig. 4J)?
3. In Fig. 5K, the authors showed delayed growth of contralateral M38 tumor in 10Gy+aCSF1 treated animals but not 10Gy+IgG treated animals. It will be interesting to explore and compare the macrophage and CD8 T cell phenotypes in contralateral and ipsilateral tumors in these mice.

Minor comments:

1. In general, the figure legends are poorly written and some figures are not well labeled, which makes it very difficult to read. Legends of Fig. 1 E-H, Fig. 2 I-J, Fig. 3 B-E and Fig. 6 C-D do not match figures.
2. The authors should specify the type of tumor (MC38 or KPC) in the legends of Fig. 4A-B.
3. The authors should clarify what "control" means in each figure legend. For example, is the control in Fig. 3H-I unirradiated animal treated with IgG? Is the control in Fig. 6I animal treated with 10Gy? What about the control Fig. 1D?
4. Color scheme of Fig. 2G should be consistent with other figures, which is, blue bar for control and red bar for 10Gy.
5. Title of Fig. 6 needs to be rephrased.

1st Revision - authors' response

12 September 2018

Referee #1 (Remarks for Author):

Combination therapies involving radiotherapy and checkpoint inhibitors, or CFS-1/CFS-1R have demonstrated activity in multiple tumour types, in clinical and pre-clinical studies, therefore although the triple combination has not been explored yet, the study is not totally innovative.

In the first paragraph of "Results", describing the results of FACS analysis the authors state that the number of macrophages, as well as of other immune subsets increases following irradiation. Since FACS analysis gives relative and not absolute results the word number should be changed in percentages and the authors should more carefully comment the results as increased infiltration. There is also another possibility i.e. that irradiation kills more tumor cells than immune cells, and that especially macrophages, which are not highly proliferating cells, are more spared than tumor cells and their percentage is relatively increased.

We agree that making this distinction is important. We have therefore removed any reference to the absolute number of immune cells. Instead, we now refer to relative changes or infiltration (highlighted throughout, paragraph 2, page 4).

In response to the second point, the single dose of irradiation is maximally lethal to tumour cells within the first 48 hours. However we see a continued increase in macrophage infiltration up to 5 days following IR. This would suggest that the relative number of macrophages continues to increase despite a reduction in the rate of tumour cell death.

Legend to Fig. 1 has some mistakes in the indications of panels E, F, G, I. In the legend of panel F it is written that "Dashed lines reflect mean TAMs within control tumours", while in the figure there are no dashed lines.

This has been amended. Lines are now solid colours, with corresponding referencing in the figure legend (highlighted).

Fig. 2. In Fig. 2A While in KPC cells the histogram plot show two clear peaks representing iNOS positive and negative cells, there is only one peak in MC38 tumors (corresponding to isotype),

showing that there are no iNOS expressing macrophages neither in the control nor after irradiation. I therefore cannot understand where the percentages of iNOS positive macrophages shown in the graph come from. Starting from this observation, strong doubts arise about the reliability of the graphs shown in Fig. 2D and 2F (ratios of M1/M2 macrophages).

The MC38 tumours contain significantly fewer TAMs compared with the KPC tumours. Consequently, the number of TAMs analysed in the macrophage gate was smaller, and the small number of iNOS positive cells did not appear as a significant peak. This is also compounded by the fact that the iNOS+ TAMs in the MC38 tumours had lower iNOS signal, compared with higher signal in iNOS+ TAMs in KPC tumours. Whilst the iNOS+ TAMs were positive in both cases (based upon the isotype control), KPC TAMs demonstrated higher signal. We have now made reference to these observations in the text (highlighted, page 4, paragraph 3). We have also changed the histogram plot presented in figure 2A to one with a more prominent peak, thus making it easier for the reader to interpret. We have also added solid lines to indicate the transition point between negative and positive signal based on the isotype control.

There are also some concerns regarding the comment to Fig. 2, the authors state that irradiation leads to increased T cell suppression (Figure 2 I,J). Nevertheless, in the case of T cells incubated with MC38-derived TAM after irradiation, the cell proliferation is significantly increased with respect to non-irradiated MC38-derived TAM.

TAMs from both irradiated MC38 and KPC tumours were suppressive. The difference lies in that TAMs from naïve MC38 tumours were also suppressive. The re-analysis of the data using the non-parametric Kruskal-Wallis test reduced the difference between TAMs from irradiated and naïve MC38 tumours to non-significant. The text has been altered to clarify the point made by the reviewer (highlighted, page 4, paragraph 3).

Fig. 4. The authors' statement that increased T cell numbers are observed after anti-CSF treatment should be rephrased and contemplate the possibility that the percentage of T cells is relatively increased with respect to macrophages which are indeed diminished by the treatment.

We agree and have changed the wording in the manuscript text to highlight that the relative increase in T cells corresponds with the reduction in TAMs associated with aCSF treatment (highlighted, page 5 paragraphs 3 and 4). Of note, in KPC tumours, where we also see a significant reduction in TAMs following aCSF, there is no increase in CD8 numbers. This finding is not in keeping with the possibility that the CD8 changes are simply due to a reduced total cell count.

In the tumor growth experiments shown in Fig. 4I 4J 4L 4M, it would be advisable to show also the growth of untreated tumors.

These groups (isotype control treated) have now been added to the graphs (Figure 4I 4J 4L 4M).

Fig. 6. In the tumor growth experiments one control is lacking. The authors should show the tumor growth after treatment with anti-PD-L1 and anti-CSF. It appears to me that in MC38 tumours there is no additive antitumor effect in adding a-PD-L1 to the combination of irradiation and anti-CSF. The appropriate control is needed to understand whether the triple combination is necessary or if anti-PD-L1 and anti-CSF are effective in double combination.

As suggested by the reviewer, we have added growth data for groups receiving antibody treatments alone (IgG, aCSF, aPDL1) or in combination (aCSF + aPDL1), see Figure 6 G,H. These experiments were performed as arms of the experiments presented in the initial submission, however they were not included in the original submission with the intention of simplifying the figures. The reviewer is correct in the observation that the addition of aPDL1 did not augment the antitumour effect. Treatment with aCSF and aPDL1 without radiation (double combination) did not result in any discernible antitumour effect.

The statistical analysis is not convincing throughout the whole manuscript. In fact, most results have been analyzed by a parametric test (i.e. T-test) but there is no mention of the analysis of the normal distribution of data. A non-parametric test should be used.

In response to this observation we have re-analysed all of the data presented in the manuscript. Briefly, to determine normality, data were analysed by the D'Agostino Pearson Omnibus normality test. For parametric data, two-tailed unpaired Student's t-test (2 groups), and one-way ANOVA with Bonferroni correction (>2 groups) were used. For non-parametric data, Mann-Whitney (2 groups) and the Kruskal-Wallis (>2 groups) with Dunn's multiple comparisons were used. This approach has been outlined in a new paragraph in the methods section ('Statistical analysis' – highlighted, page 12, paragraph 1). We have also indicated which test was used in each figure legend, and highlighted this where it has changed. A summary of any statistical significance which has altered following the new analysis is presented below.

Fig. 1B – Significance at 24 hours = <0.05 (previously <0.01)

Fig. 1D - Significance at D3 hours = <0.05 (previously <0.01)

Fig. 2A – Significance for 10Gy group = <0.05 (previously <0.01)

Fig. 2F – Significance for 10Gy group = <0.01 (previously <0.001)

*Fig. 2I – Significance for the difference between TAMs from non-irradiated vs irradiated tumours is now non-significant.

- Significance for the 10Gy TAM group = <0.05 (previously <0.001)

Fig. 2J – Significance TAMs from irradiated tumour group = <0.05 (previously <0.01)

Fig. 6A – Significance for 10Gy group = <0.05 (previously <0.01)

***Only in Figure 2I did re-analysis reveal results which altered significance from those presented in the original submission.**

Referee #2 (Remarks for Author):

In this manuscript, the authors investigated the function of macrophages in tumours treated by irradiation. They found that there was increased CSF1 secretion by tumour cells upon irradiation and there was also a concurrent increase of macrophages in irradiated tumours which displayed an immunosuppressive phenotype. When combined with radiation, macrophage depletion by anti-CSF1 delayed tumour growth compared to animals received radiation only. And this delay was abolished by depleting CD8 T cells via anti-CD8. Moreover, radiation induced higher PD-L1 expression on tumour cells. Combined treatment with irradiation + systemic aPD-L1 + aCSF1 led to tumour regression in 3 out of 8 KPC tumours, which are highly resistant to irradiation or aPD-L1 treatment alone.

Overall, the authors showed that irradiation leads to a highly immunosuppressive tumour microenvironment by recruiting tumour associated macrophages and inducing PD-L1 expression in tumour cells. Importantly, combined therapy showed promising tumour regression phenotype in mouse KPC tumours. The authors need to address the following issues:

Main comments:

1. The authors stated that "CD8 completely abrogated the tumour growth delay observed in previous experiments" but data shown in Fig.4I, J, L, M was not sufficient to show the abolition. The authors should include tumour growth curves of animals treated with (1) IgG only, (2) 10Gy+IgG and (3) 10Gy+aCSF.

These groups have now been added to the graphs. These experiments were performed as arms of the experiments presented in the initial submission, however they were not included in the original submission with the intention of simplifying the figures.

2. Given that there was little CD8 T cell infiltration in KPC as shown in Fig. 4B and H. Could the authors comment on why the delayed tumour growth in aCSF treated KPC mice was abrogated by aCD8 treatment (Fig. 4J)?

In the absence of an absolute change in the quantity of CD8 T cells, previous literature indicates that the spatial distribution within a tumour may also be important for antitumour responses⁽¹⁻³⁾. We therefore performed immunofluorescent staining for CD8 cells in KPC tumours (Fig 4H). We noted that there were very few CD8 T cells present within the centre of tumours treated with 10Gy irradiation alone. After treatment with aCSF however, there appeared to be more CD8 T cells

present in the centre of the tumour. Previous studies have identified tumours with an ‘immune excluded’ phenotype, characterised by the accumulation of lymphocytes on the periphery of tumours. We have referenced these articles in the discussion (highlighted, page 9, paragraph 2).

3. In Fig. 5K, the authors showed delayed growth of contralateral M38 tumour in 10Gy+aCSF1 treated animals but not 10Gy+IgG treated animals. It will be interesting to explore and compare the macrophage and CD8 T cell phenotypes in contralateral and ipsilateral tumours in these mice.

We have repeated the experiments and analysed the TAM and CD8 populations in the contralateral tumours. In summary, the response in the primary (index) tumours was comparable to the response observed in our previous experiments. However in the secondary tumours, aCSF was less effective at reducing TAMs when the primary tumour was irradiated. Whilst the number of CD8 T cells was increased in the secondary tumour, this was to a lesser extent compared with treatment with aCSF alone. This is possibly due to the less significant TAM depletion. These results have been included in Figure 5, N-O. We have also commented on the results in the results section (highlighted, page 7, paragraph 1). We have also changed the nomenclature to call the irradiated or the ipsilateral tumour the primary and the contralateral the secondary.

Minor comments:

1. In general, the figure legends are poorly written and some figures are not well labelled, which makes it very difficult to read. Legends of Fig.1 E-H, Fig.2 I-J, Fig.3 B-E and Fig.6 C-D do not match figures.

We have changed the figure legends throughout, to make them clearer and easier to read. These changes have been highlighted. The legends for Figure 1 E-H have been changed to match the figure.

2. The authors should specify the type of tumour (MC38 or KPC) in the legends of Fig. 4A-B.

We have clarified the cell line in the figure legend (highlighted).

3. The authors should clarify what "control" means in each figure legend. For example, is the control in Fig.3H-I unirradiated animal treated with IgG? Is the control in Fig. 6I animal treated with 10Gy? What about the control Fig. 1D?

In order to avoid confusion, we have changed any reference to ‘control’ throughout the figures and corresponding legends. Where experiments only include samples which have been mock-irradiated, they are now labelled as ‘0Gy’. For antibody experiments, groups receiving isotype antibodies are now referred to as ‘IgG’.

4. Colour scheme of Fig.2G should be consistent with other figures, which is, blue bar for control and red bar for 10Gy.

Figure 2 G in our submitted copy was colour coded. The panel below (Fig 2I-J) was not however, and we have changed this.

5. Title of Fig.6 needs to be rephrased.

This has been rephrased (highlighted in Figure Legends).

1. Galon, J. *et al.* Type, Density, and Location of Immune Cells Within Human Colorectal Tumors Predict Clinical Outcome. *Science* (80-.). **313**, (2006).
2. Berthel, A. *et al.* Detailed resolution analysis reveals spatial T cell heterogeneity in the invasive margin of colorectal cancer liver metastases associated with improved survival. (2017). doi:10.1080/2162402X.2017.1286436
3. Chen, D. S. & Mellman, I. Elements of cancer immunity and the cancer-immune set point. *Nature* **541**, 321–330 (2017).

Thank you for the submission of your revised manuscript to EMBO Molecular Medicine. We have now received the enclosed reports from the referees that were asked to re-assess it. As you will see the reviewers are now globally supportive and I am pleased to inform you that we will be able to accept your manuscript pending the following final editorial amendments.

***** Reviewer's comments *****

Referee #1 (Remarks for Author):

I believe that the authors improved the quality of figures, text and statistical analysis. As far as I'm concerned, this new version of the manuscript is suitable for publication.

Referee #2 (Remarks for Author):

Most of my concerns have been addressed.

However, I still have major concerns regarding the authors' interpretation of the double tumor model (Fig. 5 N-O). The authors stated that 'systemic response to irradiation of the primary tumour causes increased TAMs in the secondary tumour, limiting CD8 infiltration'. This statement is misleading. Because TAM percentage (%CD11b+F4/80+) in the IgG group and 10Gy+IgG group of MC38 secondary tumors showed no significant difference (Fig. 5N). It seems that irradiation did not increase TAMs in the secondary tumors, but somehow rendered TAMs more resistant to aCSF1. Also, when comparing the 10Gy+IgG group and 10Gy+aCSF1 group, one notices that despite the decrease of TAMs after the addition of aCSF1, there was no increase of CD8 T cells. Could the authors comment on this?

Corresponding Author Name: Ruth Muschel
Journal Submitted to: EMBO Molecular Medicine
Manuscript Number: EMM-2018-09342